# How to Optimize Cardioversion of Atrial Fibrillation

**DOI:** 10.3390/jcm11123372

**Published:** 2022-06-12

**Authors:** K. E. Juhani Airaksinen

**Affiliations:** Heart Center, Turku University Hospital and University of Turku, 20521 Turku, Finland; juhani.airaksinen@tyks.fi; Tel.: +358-405876052

**Keywords:** antiarrhythmic drugs, anticoagulation, atrial fibrillation, cardioversion, rhythm control, stroke, thromboembolic complication

## Abstract

Cardioversion (CV) is an essential component of rhythm control strategy in the treatment of atrial fibrillation (AF). Timing of CV is an important manageable factor in optimizing the safety and efficacy of CV. Based on observational studies, the success rate of CV seems to be best (≈95%) at 12–48 h after the onset of arrhythmic symptoms compared with a lower success rate of ≈85% in later elective CV. Early AF recurrences are also less common after acute CV compared with later elective CV. CV causes a temporary increase in the risk of thromboembolic complications. Effective anticoagulation reduces this risk, especially during the first 2 weeks after successful CV. However, even during therapeutic anticoagulation, each elective CV increases the risk of stroke 4-fold (0.4% vs. 0.1%) during the first month after the procedure, compared with acute (<48 h) CV or avoiding CV. Spontaneous CVs are common during the early hours of AF. The short wait-and-see approach, up to 24–48 h, is a reasonable option for otherwise healthy but mildly symptomatic patients who are using therapeutic anticoagulation, since they are most likely to have spontaneous rhythm conversion and have no need for active CV. The probability of early treatment failure and antiarrhythmic treatment options should be evaluated before proceeding to CV to avoid the risks of futile CVs.

## 1. Introduction

Cardioversion (CV) is an essential component of the rhythm control strategy for atrial fibrillation (AF). Traditionally, the procedure has been classified as acute CV when performed within 48 h from the onset of arrhythmic symptoms and elective CV when performed later after the initiation of therapeutic anticoagulation. Successful CV delivers instantaneous relief from arrhythmic symptoms and is often the desired treatment option from the patient’s perspective. Based on the currently available evidence, the main motivation for pursuing the rhythm control strategy is to reduce AF related symptoms and improve quality of life. Antiarrhythmic drugs and catheter ablation are often needed to maintain sinus rhythm after successful CV [1,2]. A recent study showed that an early initiation of rhythm control therapies is associated with less frequent cardiovascular complications, but more often causes other adverse events [3]. The main drawback of CV is the temporary increase in the risk of thromboembolic events after rhythm conversion [4,5]. Thus, the probability of successful rhythm conversion and maintenance of sinus rhythm should be always weighed against the safety of CV, such as the risks of thromboembolic and acute arrhythmic complications of CV. This clinical review focuses on electrical CV and gives a short overview of measures that can optimize the efficacy and safety of CV in everyday clinical practice.

## 2. How to Improve the Success of CV

The timing of CV is the most significant predictor of successful rhythm conversion [6]. The success rate of electrical CV performed within 48 h from the onset of arrhythmic symptoms has ranged from 85% to 97% and during the last decade from 92% to 97% [6,7,8,9]. The success of elective CV is lower, varying from 66% to 95% [6,7,10]. In the 2012 Euro Heart Survey, the success rate of acute CV was higher when compared to elective CV of longer arrhythmias (97% vs. 87%, *p* = 0.003) [7]. Similarly, in the FinCV studies, the success rate was 94.5% in 7660 CVs performed within 48 h from the symptom onset and 84.8% in the elective CVs (*N* = 1998) [8,11]. Prolonged arrhythmia may cause electrical and structural remodeling and CV failures become more common when the duration of persistent AF is prolonged [7,10,12]. At the other end of the spectrum, the strong autonomic arousal and ongoing arrhythmic triggers may compromise the CV’s success during the very early hours of arrhythmia [8]. Thus, when targeting a high initial CV success rate, the optimal timing of CV seems to be 12–48 h after the onset of arrhythmic symptoms [11].

In the acute (<48 h) setting, pharmacological CV is a reasonable alternative to electrical CV, since it does not require fasting and anesthesia. Flecainide, propafenone and vernakalant are the currently used drugs in recent-onset AF. These drugs are safe in patients without structural heart disease, but the efficacy of pharmacological CV is less satisfactory than electrical CV [2,9,13]. There is some evidence that the success rate is somewhat higher (around 70%) among patients with a short (<12 h) AF, but the frequent spontaneous CVs are probably contributing to the better efficacy of the procedure during the early hours of AF symptoms [2,14].

Other measures which improve the success rate of electrical CV, especially in persistent AF or in obese patients, include starting directly with high-energy (200 J) shocks and using handheld paddles or pressure upon adhesive patches together with biphasic defibrillators capable of delivering shocks up to 360 J [15,16]. Pre-treatment with antiarrhythmic drugs—most often amiodarone—is a widespread practice for elective CV and increases the likelihood of restoring and maintaining sinus rhythm after CV [2]. Recently, anterior-lateral electrode positioning was also shown to improve the efficacy of elective electrical CV compared with anterior-posterior positioning, but in the acute setting the difference in success rate seems insignificant [9,17].

## 3. Predictors of Early AF Recurrence after CV

In addition to the primary success of CV, evaluating the risk of early recurrence is important when deciding whether CV is to be carried out or not. The early recurrence rate after CV is much higher than the rate of primary CV failure, and new AF episodes emerge in approximately half of the patients within one year despite antiarrhythmic therapy [2,6]. Most recurrences occur early and several clinical, electrocardiographic and echocardiographic predictors for early recurrence after CV of acute AF have been identified. The recently introduced AF-CVS score utilizes five significant clinical risk factors (prior AF episodes within 30 days, older age, any previous history of AF, heart failure and vascular disease) and helps to identify patients at high (>40%) risk of unsuccessful CV or early recurrence, for example, a 66-year old patient with a history of AF and a previous AF episode within 30 days [18]. The use of this simple score may help the clinician to decide upon the best treatment strategy for patients at high risk of initial CV failure and early AF recurrence.

Based on experimental evidence, longer AF episodes may cause AF through electrophysiological mechanisms [19]. In line with this background, the early clinical recurrence rate is higher after elective CV than after CV of acute AF. In the FinCV studies, the 30 day AF recurrence rate after acute CV was 17.3% and 32.4% after elective CV [11]. Numerous weak and inconsistent clinical predictors for early recurrence after elective CV have been identified. Echocardiographic and electrocardiographic features reflecting atrial pathology (e.g., increased left atrial size, low left atrial appendage flow velocity, advanced interatrial block) predict AF recurrence, but when the features are not advanced, they are not strong enough for clinical decision-making [20]. Frequent use of antiarrhythmic drugs and catheter ablation therapies, uncertain duration of AF, and vague symptoms render the analysis of risk factors for recurrences after elective CV difficult. There is, however, growing evidence to support that the longer duration of persistent arrhythmia prior to elective CV is crucial for later recurrences, for example, the 1 year AF recurrence rate was lower in patients with AF duration < 3 weeks compared to those with more prolonged arrhythmia before CV (41.1% vs. 57.9%, *p* < 0.01) [21]. Similarly, in the X-VeRT trial, patients receiving early CV (<6 days after hospitalization) had a 67% higher probability to regain sinus rhythm at the end of the study than those with delayed CV between 21 and 56 days after hospitalization [22]. Based on this clinical and experimental evidence, unnecessary delaying of elective CVs seems unreasonable [6].

## 4. How to Minimize the Risk of Thromboembolic Complications after CV

The risk of thromboembolic complications after CV is well-established and the mechanisms behind the risk have been comprehensively described in recent reviews [5,13]. Up to 6.4% of all ischemic strokes in patients with paroxysmal/persistent AF were preceded by CV within 30 days in a recent analysis [23].

The risk of stroke after elective CV has decreased from the early reports of 3.4–7% to below 1% with the use of vitamin K antagonists [4,5]. Current guidelines recommend effective anticoagulation for three weeks before a scheduled CV and for at least four weeks afterwards, and permanently for patients with stroke risk factors [2]. Direct oral anticoagulants are preferred, since the transient prothrombotic effects during the initiation of vitamin K antagonists together with more labile anticoagulation may increase the thromboembolic risk. In a meta-analysis combining data from randomized trials, the incidence of thromboembolic events within 30 days of CV was 0.41% when using direct oral anticoagulants with no statistically significant difference to vitamin K antagonists (0.61%) [24]. The risk seems low at first glance, but in the randomized trials (RE-LY, ARISTOTLE, ENGAGE-AF, ROCKET-AF) the rate of ischemic strokes has varied between 0.08% and 0.12% per month, suggesting that each elective CV—even when performed during therapeutic anticoagulation—predisposes the patient to a 4-fold increased risk of stroke during the first month after the procedure (0.4% vs. 0.1%) [5].

The incidence of thromboembolic complications after CV performed for acute (<48 h) AF without anticoagulation has ranged from 0% to 0.9% (mean = 0.7%) and most of the strokes occur during the first week after CV [5,13]. The individual stroke risk varies, however, considerably (up to 9.8%) according to the risk factors and the CHA_2_DS_2_VASc–score, which is used to predict thromboembolic risk in this scenario [25,26]. The CV delay is of importance even in this early setting; the thromboembolic risk is low (0.3%) when CV is performed within 12 h from the symptom onset, compared with the 4-fold risk of 1.2% with a longer delay of 12–48 h [27]. This finding is not unexpected, since prothrombotic changes develop early, and more comprehensive coagulation abnormalities are observed already at 12 h after AF onset [5]. In low risk (CHA_2_DS_2_VASc < 2) patients without ongoing anticoagulation, it seems rational to perform CV within 12 h of AF onset to minimize the risk of thromboembolism. With this strategy, it is possible to avoid the bleeding risk caused by initiation of short-term anticoagulation [28].

The risk of thromboembolic complications after acute (<48 h) CV is low during oral anticoagulation; 0.1% in the FinCV Study (2298 CVs) [26]. Thus, it seems reasonable to aim at early (within 48 h of AF onset) CV also in patients on long-term oral anticoagulation, since the overall risk of post-CV thromboembolic complications is numerically lower (0.1% vs. 0.41–0.61%) than in later elective CV performed during therapeutic oral anticoagulation [29-31].

Effective anticoagulation is of utmost importance at the time of CV, and especially during the first days after CV, since the risk of stroke is highest during this vulnerable period of atrial stunning and >80% of strokes occur within the first week after the procedure [13]. Successful rhythm conversion may create a false impression of safety for the patient and lead to harmful interruptions in anticoagulation. Hellman et al. observed subtherapeutic (<2) INR values in nearly a quarter of patients within 3 weeks after CV and this drop was related to a higher (1.7% vs. 0.3%, *p* = 0.03) risk of thromboembolic events compared with the patients having stable therapeutic anticoagulation [32]. Knowing the compliance problems with direct oral anticoagulants, patients’ adherence to the prescribed anticoagulants is particularly important during this vulnerable period, as there is a strong correlation between the increased risk of stroke and the reduction of anticoagulant use. It is tempting to speculate—but almost impossible to show—that intensified anticoagulation during the first high-risk days after CV might help to reduce stroke risk in patients with high CHA_2_DS_2_VASc score. In EMANATE study, no thromboembolic events were seen in 331 patients receiving a 10 mg loading dose of apixaban [30]. Similarly, in two studies on elective CV, the risk of stroke was low (0.0–0.1%) in 1674 patients with INR ≥ 2.5 during CV [32,33].

Uncertain AF duration combined with inadequate anticoagulation is an important indication to postpone CV, and to instead institute effective anticoagulation for three weeks before a later elective CV. If early CV is desirable, current guidelines recommend transesophageal echocardiography to exclude intracardiac thrombi [2]. In a recent meta-analysis, the pooled prevalence of left atrial thrombus in patients who were not on oral anticoagulation but were undergoing CV or catheter ablation was 1.8%; and the prevalence was significantly higher among patients with non-paroxysmal AF [34]. The main shortcoming of transesophageal echocardiography is that it only detects the pre-existing thrombi in the left atrium or left atrial appendage, and it cannot protect from thromboembolic complications caused by thrombi that are formed later, during the stunning phase after CV. Importantly, the absence of intracardiac thrombus in transesophageal echocardiography does not decrease the need for effective anticoagulation after CV [13]. On the other hand, if a thrombus is detected, CV should be postponed and therapeutic anticoagulation started.

## 5. Incidence of Arrhythmic Complications after CV

Immediate arrhythmic complications after electrical CV of AF are quite rare. In the 2012 Euro Heart Survey of 712 electrical CVs, the rate of ventricular fibrillation, ventricular tachycardia and Torsades des Pointes was 0.4%, 0.8% and 0.1%, respectively [7]. On the other hand, no such ventricular arrhythmias were found in the FinCV studies including 10,852 electrical CVs, or in 543 elective CVs reported by Morani et al. [11,35,36]. Randomized trials assessing direct oral anticoagulants during CV have not reported on ventricular arrhythmias.

The incidence of bradycardic complications has varied from 0% to 0.9% of the electrical CVs performed for acute AF [6]. Most of these bradyarrhythmias have been transient with no need for specific treatment. The rate of bradyarrhythmic complications after elective CV of persistent AF has ranged from 0.8% to 1.5% suggesting that the longer duration of AF may lead to a minor increase in the risk of bradycardic episodes after CV [11,33]. It seems that most bradycardic complications reflect underlying sinus node dysfunction and become more common with advanced age, at least in acute AF. Female gender may also predispose to bradycardic events. The incidence of bradyarrhythmias was markedly increased (~3%) among female patients > 75 years of age in the FinCV study, but contrary to common beliefs, slow ventricular rate of AF and the use of beta blockers or digoxin was not associated with a higher risk for asystole or bradycardia [35].

## 6. How to Avoid Futile CVs

Rational patient selection and timing of CV are the key elements for the optimal use of CV in the rhythm control strategy. The decision to proceed with CV depends on hemodynamic status and severity of arrhythmic symptoms together with the risks of rhythm control strategy. Early AF recurrences are common, and antiarrhythmic treatment options should be evaluated in patients with mild arrhythmic symptoms who are at a high risk of early recurrences (high AF-CVS score) before proceeding with CV to avoid the risks of repeated futile CVs.

Spontaneous rhythm conversion is common during the early hours of acute AF occurring in 32–73% of patients within 24 h and in 52–77% during an observation period of up to 48 h [37]. Absence of any underlying heart disease or left atrial enlargement, age < 60 years, AF duration < 24 h, and the patient having no history of persistent AF are the most consistent predictors of spontaneous CV [37]. The RACE 7 ACWAS trial showed that a wait-and-see approach (<48 h) CV was non-inferior to early CV in providing sinus rhythm at 4 weeks, and almost 70% of patients with recent-onset AF regained sinus rhythm spontaneously while waiting [38]. In view of the above findings, the wait-and-see approach (24–48 h) is a reasonable option for otherwise healthy patients using oral anticoagulation when their arrhythmic symptoms are mild. It is important to shorten the emergency department visit and it is reasonable to advise patients with recurrent AF episodes to wait the early hours of arrhythmia at home if their symptoms are well-tolerated and if they are using effective oral anticoagulation. If needed, patients can be taught to use rate control medications (beta blockers or verapamil) to alleviate palpitations. Some of these patients may already have experienced previous self-termination of arrhythmia, thus reassuring the relevance of this approach. Pill-in-the-pocket approach is also an option for selected patients to reduce emergency department visits. Treatment failures and adverse events limit the wider use of this strategy.

## 7. Conclusions

Hemodynamic compromise or incapacitating symptoms indicate the need for urgent CV. The risk of stroke is the main concern when undertaking this procedure, and the magnitude of this risk increases by delaying it. Effective anticoagulation (preferably using direct oral anticoagulants), especially during the early days after CV, prevents most of the thromboembolic complications. Acute CV can be performed without anticoagulation only in low-risk patients and preferably within 12 hours of AF onset. Short waiting for spontaneous CVs, which are common during the early hours of arrhythmia, helps to reduce the need of active CVs in selected patients with milder arrhythmic symptoms. Compared with later elective CV, early (<48 h) CV more often restores sinus rhythm and may also reduce the risk of thromboembolism during ongoing anticoagulation therapy. The optimal timing of CV seems to be 12–48 h after the onset of arrhythmic symptoms in the majority of patients. Patients at high risk of early recurrences after successful CV should be identified and their antiarrhythmic treatment options should be evaluated before proceeding with CV to avoid the risks of futile CVs.

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
