# Peer review of "How to Optimize Cardioversion of Atrial Fibrillation"

_jcm, 2022, doi:10.3390/jcm11123372_

Round 1
Reviewer 1 Report
How to optimize cardioversion of atrial fibrillation/ reviewer comments
In this review, Airaksinen et al try to summarize the different aspects involved in AF cardioversion, with the aim of recommending the best approach regarding each of these aspects.
Follow are my major comments:
- The authors should make the definitions of acute versus elective DCCV from the first paragraph of the introduction.
- In the first section of how to optimize the success of AF DCCV I would expect to see another long paragraph discussing in detail the various anti-arrhythmic drags (AAD) which could potentiate success of AF DCCV, reviewing the literature for those AAD which were shown to be superior to others regarding DCCV procedure success, and which AAD's are best for maintaining sinus rhythm after the resumption of sinus rhythm. As of now, there is only one sentence dedicated for these drugs and this is definitely lacking.
- The section of predictors of early AF recurrence is also lacking. I would like to see more risk factors associated with AF recurrence which are not mentioned. The authors should do better work here, and not just describe a 5 risk factors score. There are definitely much more factors associated with AF recurrence post DCCV (just a few of these are obstructive sleep apnea, LA size, etc…) Moreover, they should show the various literature describing all CXHADVASC risk factors for stroke to be associated with AF recurrence as well. Indeed, this paragraph needs a significant extension and improvement.
- Regarding section 3 of thromboembolic risk- there are a few comments:
a) In the 1st paragraph the authors say that 6/4% of all strokes related to AF occur within a month from DCCV. However, they should mention large studies which did not show an exact time relation between AF documentation and stroke, suggesting that the stroke mechanism is not definitely mechanistically but might relate AF risk factors (to the point that some argue that stroke does not correlate with AF perse but rather to AF vascular risk factors.
b) In the 2nd paragraph the authors recommend early DCCV within 12 hours to avoid bleeding risk associated with anti-coagulation. However, the guidelines clearly say that in some patients anti-coagulation should always be given, for example- patients post thromboembolic event, rheumatic mitral stenosis, and with mechanical valves.
- Regarding section 4 of post DCCV arrhythmic complications: I think this section should be extended to include multiple trials related to drug-induced bradycardia among AF patients in general and after DCCV specifically. For example, multiple studies showed a significant association between pre-treatment by Amiodarone and post DCCV bradycardia. Moreover, in meta-analysis the combination of amiodarone with Digoxin was the best predictor for post DCCV bradycardia.
- In section 5, the authors mention the RACE7 study advocating possibility of wait and see approach in relatively young patients who are effectively anti-coagulated, revealing similar 30-day sinus rhythm incidence. Nevertheless, the authors should discuss the potential disadvantages of waiting in such cases, given the multiple data in their prior sections, implying the advantages of really early (<12 hours) AF cardioversion 9ex. Very low incidence of thromboembolic phenomena, best DCCV success, etc…)
Author Response
Please see below the attachments.

Reviewer 2 Report
he Review How to Optimize Cardioversion of Atrial Fibrillation by Juhani Airaksinen is
A Comprehensive sate of the art but Author in reviewer’s opinion has missed some important issues.
Paragraph 2. Predictors of early AF recurrence after CV. Author has focused on time periods (acute or persistence AF) as predictors of early AF recurrence after CV. There is lack of information about morphological and clinical predictors as enlargement of the left atrium (e.g. LA area or LAVI) as well as residual flow velocity of LA appendage. Also impact of mitral valve comorbidities as regurgitation and stenosis is with to take into consideration. Is there any data regarding drug intake before elective CV on (e.g. amiodarone) AF recurrence after CV?
Paragraph 3. Fully agree the TEE does not predict thrombus formation but on the other hand the information regarding thrombus in LAA despite adequate anticoagulation therapy is crucial and is contraindication for CV. What is the author opinion about performing TEE before all CV? Finally what author would recommend how to menage patients with LAA thrombus and adequate anticoagulation therapy ?
Technical remarks:
Different fonts have been used
The sentence not correctly divided (189/190)
Author Response
Please see below.
